# Obj2Seq: Formatting Objects as Sequences with Class Prompt for Visual Tasks

**Zhiyang Chen**[1,2] **Yousong Zhu**[1] **Zhaowen Li**[1,2] **Fan Yang**[1,3] **Wei Li**[4]
**Haixin Wang**[1,2] **Chaoyang Zhao**[1] **Liwei Wu**[4] **Rui Zhao**[4] **Jinqiao Wang**[1,2,3]
**Ming Tang**[1]

[1]National Laboratory of Pattern Recognition, Institute of Automation,
Chinese Academy of Sciences
[2]School of Artificial Intelligence, University of Chinese Academy of Sciences
[3]Peng Cheng Laboratory
[4]SenseTime Research
{zhiyang.chen,yousong.zhu,zhaowen.li,haixin.wang}@nlpr.ia.ac.cn
yangfan_2022@ia.ac.cn  {liwei1,wuliwei,zhaorui}@sensetime.com
{chaoyang.zhao,tangm,jqwang}@nlpr.ia.ac.cn

## Abstract

Visual tasks vary a lot in their output formats and concerned contents, therefore it is hard to process them with an identical structure. One main obstacle lies in the high-dimensional outputs in object-level visual tasks. In this paper, we propose an object-centric vision framework, Obj2Seq. Obj2Seq takes objects as basic units, and regards most object-level visual tasks as sequence generation problems of objects. Therefore, these visual tasks can be decoupled into two steps. First recognize objects of given categories, and then generate a sequence for each of these objects. The definition of the output sequences varies for different tasks, and the model is supervised by matching these sequences with ground-truth targets. Obj2Seq is able to flexibly determine input categories to satisfy customized requirements, and be easily extended to different visual tasks. When experimenting on MS COCO, Obj2Seq achieves 45.7% AP on object detection, 89.0% AP on multi-label classification and 65.0% AP on human pose estimation. These results demonstrate its potential to be generally applied to different visual tasks. Code has been made available at: https://github.com/CASIA-IVA-Lab/Obj2Seq.

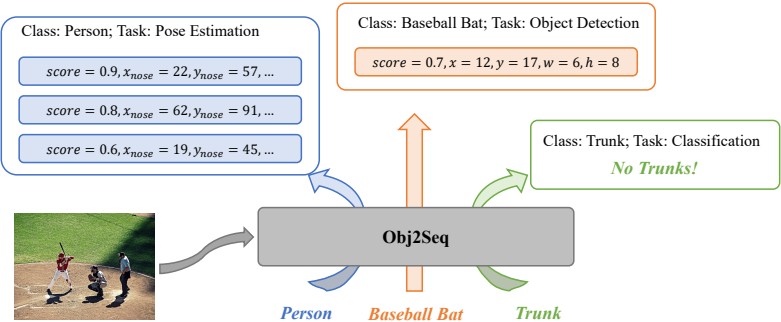

Figure 1: Obj2Seq is a unified framework for visual tasks. It takes categories as inputs, and generates desired output sequences according to different tasks.

36th Conference on Neural Information Processing Systems (NeurIPS 2022).

# 1 Introduction

Deep learning has made great progress in the field of computer vision [20, 35, 19, 26], and recently researchers are seeking for a unified framework for various visual tasks. This idea has been widely discussed in Natural Language Processing (NLP), which results in the emergence of general language models. In NLP, all tasks are formatted as sequence-to-sequence problems in order to be processed by a single transformer [43]. Meanwhile, prompt learning [16] is utilized so that the general model is compelled to perform a specific task. With the help of large-scale text data, numerous pre-trained language models are developed [13, 1, 49], and they achieve excellent performance on a wide range of language tasks.

With these encouraging achievements in NLP, it is natural to ask whether we can implement a similar framework in computer vision as well. One of the two obstacles to realize this is that visual tasks are defined in various formats. They can hardly be unified like sequence-to-sequence problems in NLP, especially those object-level tasks with high-dimensional outputs. The other is that the model may encounter diverse categories even for an identical task. If the model is not aware of what it will process in advance, we cannot expect it to satisfy customized requirements flexibly. There are some works taking a step forward to solve these problems. M-DETR [23] takes sentences as inputs in addition to the images, so that it is able to flexibly detect specific targets that mentioned and realize target-aware inference. However, M-DETR is designed for object detection only, and hard to extend for general usage. Pix2Seq [9] is a forward looking work to unify the formats of different tasks. It turns object detection into a sequence generation task, and directly applies next token generation to yield bounding boxes and categories. Even so, Pix2Seq is not aware of the desired targets before inference, and the output sequence might become extremely long under sophisticated scenes.

In this paper, we propose a unified visual framework, Obj2Seq, in order to adapt the sequence generation framework for object-level visual tasks. Obj2Seq formats these tasks into a two-step process. First, with images and class prompts as inputs, Obj2Seq determines whether each given category exists or not in an image, and detects objects of these categories. Then a sequence is generated as the output for each object in response to the desired task. Many visual tasks can be filled in this template by changing class prompts and the definition of output sequences. For each specific task, class prompts provide additional instructions to impel the framework to attend to objects of specified categories, and the output sequences are defined accordingly to include information required by the task. This gives the credit to two new modules, Prompted Visual Indicator that receives class prompts from the inputs, and General Sequence Predictor that outputs sequences in a unified format for each object. Obj2Seq has two main advantages. On one hand, the behavior of Obj2Seq can be controlled and manipulated. By taking different class prompts as inputs, Obj2Seq is able to focus on different objects and flexibly satisfy specific requirements under practical scenes. On the other hand, the sequences generated by Obj2Seq are of a unified format to describe objects, which makes it ready to be extended for a wide range of object-level visual tasks.

In order to validate the versatility of our framework, we conduct experiments on MS COCO [28] for some common visual tasks. Obj2Seq achieves 45.7% AP on object detection and 89.0% AP on multi-label classification when trained for object detection, while it also achieves 65.0% AP on human pose estimation with keypoint annotations. These results indicate that our framework is effective for different visual tasks, and has potential for general usage.

# 2 Related Works

## 2.1 General Frameworks for Visual Tasks

The idea of pretraining a general model is firstly proposed in NLP [13, 1]. Currently, a similar trend appears in computer vision as well. Researchers first borrow transformer as the base model [14, 42, 30, 45, 12], and then design novel self-supervised algorithms to pretrain them with the help of huge amounts of unlabeled data and modern hardware [10, 5, 25, 18, 8, 27]. The pretrained models are capable of extracting robust feature for downstream tasks. However, since only the pretrained backbone is provided, downstream tasks still have to design their specific head structures and need further training to get desired results.

Recently, the academic raises a higher requirement to train a general framework that can be directly applied in different tasks. It should be trained in an end-to-end way, and does not need finetuning

for a specific task. Multi Task Learning (MTL) is one related field [6, 38, 17]. It constructs a shared backbone and multiple heads to process a fixed combination of tasks. However, since MTL needs individually designed heads and sufficient labels for every sample, it is expansive and unable to be extended to other tasks. Recently, works like ML-Decoder [37] and X-DETR [2] attempt to construct a general framework for a set of some similar tasks. However, they can hardly be generalized for tasks with totally different definitions. Pix2Seq [9] provides a straight-forward solution to convert an image into a sequence of words. However, it lacks interface to flexibly adopt for different scenes. In addition, the whole image is sometimes too complex to be represented in a single sequence.

Different from previous work, we take objects as the basic unit in visual tasks, and unify them as sequence generation for each single object. Additional class prompts are used to specify the objects in need.

## 2.2 Prompt Learning

Prompt learning [16] is firstly develop in NLP in order to directly apply pre-trained language models in different tasks. It does not need finetuning model parameters, but rather modifies the inputs and uses a template to indicate the proposed task. Therefore, the model handles all tasks in a unified manner.

In order to train vision models with easily available image-text pair data, the concept of prompt is firstly introduced to construct training targets with label text in computer vision [34, 48]. These algorithms help supervision with large-scale data only for a certain task. The prompts are used as labels, and they cannot control the inference. VPT [22] and FP-DETR [44] add prompt parameters in their framework. These parameters are trained in the finetune stage for each task. However, they do not make use of language embeddings, and the effect of these modules is hard to explain. ML-Decoder [37] and MDETR [23] use text to impel the model to focus on specific categories or objects. It allows us to interfere the model behavior, however, these frameworks are limited to a group of similar tasks. In this paper, class prompts are used to give instructions about what categories to detect, and Obj2Seq is able to perform a wide range of vision tasks.

## 2.3 Sequence Prediction

Sequence prediction is a basic formulation for various NLP tasks, since the words in sentences are sequential naturally. It has the characteristic that the forward process at time $t$ depends on features or outputs from previous time steps. Currently transformer [43] is widely utilized to solve these problems. In order to generate a sequence, the output token at time $t$ is taken as the input token of the next step. Some up-to-date pre-train algorithm use the input sentence as the target output, in order to perform next-token prediction for unsupervised training [1, 49].

As to computer vision, there is no preset order in the input image. Sequence prediction is only available in image generation since they can generate an output image pixel by pixel [32]. When transformer becomes popular, it is firstly applied to process a group of image patches [14] or an unordered set of objects [4]. Recently, the idea of next-token prediction is borrowed in order to develop novel self-supervised training algorithms [7, 21]. However, these algorithms are still closely related to image generation, as they predict the image patches in sequence. Pix2Seq [9] is the first attempt to apply sequential prediction as a general method in a traditional visual task, and achieves a decent performance. It regards the whole detection results as a single sequence, which abandons semantic structures in the original image. On the contrary, our framework considers a more basic unit, objects, in the field of computer vision. This makes Obj2Seq more suitable for visual tasks, and achieves better results.

## 3 Methods

### 3.1 Overview

Obj2Seq is a unified visual framework for object-level tasks. It takes an image $\boldsymbol{I} \in \mathbb{R}^{3 \times H_0 \times W_0}$ and a group of $K$ class prompts $\boldsymbol{C} = [\boldsymbol{c}^{(1)}...\boldsymbol{c}^{(K)}] \in \mathbb{R}^{K \times d}$ as inputs, and outputs a set of objects, each with a sequence to describe it. It contains four main parts listed as below:

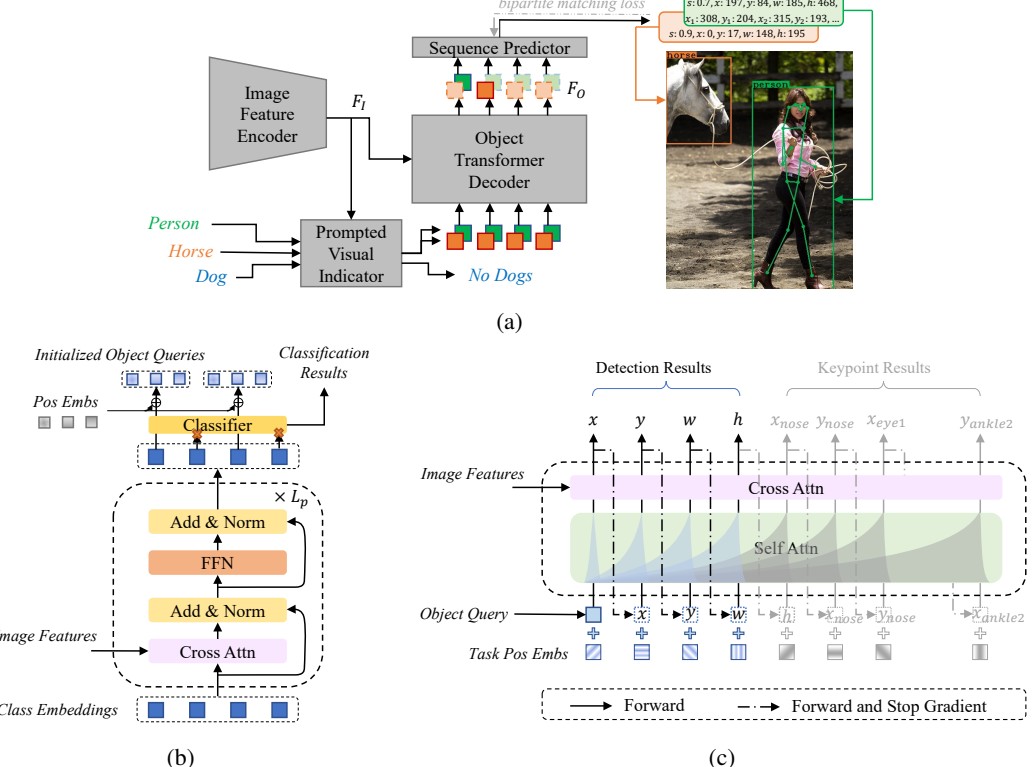

Figure 2: (a) The overall framework of Obj2Seq. (b) Illustration of Prompted Visual Indicator. (c) Illustration how General Sequence Predictor generates sequence for an object. We take object detection and human pose estimation as examples. The gray part is utilized only for human pose estimation.

- Image Feature Encoder $\mathcal{E}$ to extract image features.
- Prompted Visual Indicator $\mathcal{I}$ to receive class prompts from inputs and initialize object queries.
- Object Transformer Decoder $\mathcal{D}$ to encode object-related features into object queries.
- General Sequence Predictor $\mathcal{P}$ to generate a sequence of desired attributes for each object.

When the inputs $\boldsymbol{I}$ and $\boldsymbol{C}$ are provided, Obj2Seq first extracts image features $\boldsymbol{F}_I$ with Image Feature Encoder $\mathcal{E}$. Then, Prompted Visual Indicator $\mathcal{I}$ receives class prompts. It judges the existence of each category for classification, and initializes class-aware object queries $\hat{\boldsymbol{F}}_O$. Each object query $\hat{\boldsymbol{f}}_o \in \mathbb{R}^{1 \times d}$ is responsible for detecting and describing one object of a specific class. Afterwards, these object queries are sent into Object Transformer Decoder $\mathcal{D}$. The decoder extracts object-related features from $\boldsymbol{F}_I$ according to $\hat{\boldsymbol{F}}_O$, and transforms $\hat{\boldsymbol{F}}_O$ into well-encoded object queries $\boldsymbol{F}_O$. These new queries contain sufficient information to describe their corresponding objects as the task requires. Finally, General Sequence Predictor $\mathcal{P}$ generates a sequence for each object query, which contains the desired attributes and constitutes the final result. The whole pipeline is shown in Figure 2a.

In the following sections, we mainly elaborate on Prompted Visual Indicator $\mathcal{I}$ and General Sequence Predictor $\mathcal{P}$, and then supply additional details in other parts of our implemention.

## 3.2 Prompted Visual Indicator

Different from processing with a complete category set, human beings usually know what categories to focus on in advance of each task. This prior information makes their behavior more compact and accurate. Also, the categories can be flexibly changed according to specific requirements. Therefore,

is it possible that we provide not only the images but also additional instructions as inputs, so that the model is forced to focus on objects of specified categories?

Prompted Visual Indicator borrows the idea of prompt learning from NLP [16]. It receives additional class prompts as inputs, therefore it can classify the input image with specific categories, and generate input-dependent object queries to focus on desired objects. As shown in Figure 2b, each class prompt is a vector representing a certain category $c^{(k)} \in \mathbb{R}^d$. These input vectors are sent into prompt blocks with cross-attention modules that can extract class-related features from $\boldsymbol{F}_I$. After processing by these blocks, the output vectors $\boldsymbol{f}_c^{(k)}$ represent features about specified categories in the input image. These vectors can be classified by a class-related binary classifier as in Eq. (1). The classifier compares them with original class vectors, and outputs a score $s_C^{(k)}$ to indicate the existence of each category. Moreover, these features are utilized to generate class-aware object queries for further inference. We retain $K'$ categories that are more likely to exist with a certain retention policy, and perform Cartesian product with a set of $N$ position embeddings to generate $N$ object queries for each retained class. Each object query $\hat{\boldsymbol{f}}_o$ is the sum of a class vector $\boldsymbol{f}_c^{(k)}$ and a position embedding. These object queries are sent into Object Transformer Decoder for further process. The retention policy is detailed in Section 3.4.

$$s_C^{(k)} = sigmoid(linear(\boldsymbol{f}_c^{(k)}) \cdot \boldsymbol{c}^{(k)} / \sqrt{d}) \tag{1}$$

With the help of Prompted Visual Indicator, Obj2Seq can take not only images but also desired categories as inputs. The model can flexibly classify and detect desired objects with given categories, and therefore is able to achieve better performance. Moreover, each category is processed in an isolated way. Obj2Seq is able to take any category subsets as the input, without any influence in results. This makes our framework flexible and easy to implement in different scenes.

### 3.3 General Sequence Predictor

In the field of NLP, a task-agnostic language model usually converts all language tasks into a sequence-to-sequence format. This is intuitive, because we always answer questions by speaking a sentence out. As to visual tasks, we can also describe the input image with a sentence. Since the image contains more sophisticated information, the sentence should be better organized with a more basic visual unit, objects. In order to satisfy different requirements, we may need the bounding box, keypoint offsets or some other attributes in this sentence to describe an object. Therefore, we construct General Sequence Predictor, that generates a sequence of attributes to describe an object. This paradigm can be generalized among a wide range of visual tasks.

General Sequence Predictor $\mathcal{P}$ takes object queries processed by Object Transformer Decoder as inputs, and outputs a sequence of attributes for each object. We illustrate how it works in Figure 2c, taking object detection and human pose estimation as examples. For simplicity, we only elaborate on the prediction process for one object query, and omit the superscript $(i)$. The predictor $\mathcal{P}$ first takes the object query $\boldsymbol{f}_o$ as the input $\boldsymbol{z}_{in,0}$ of the first time step. The input is then sent into self- and cross-attention layers to get the output feature $\boldsymbol{z}_{out,0}$. For any other time step $t$, $\mathcal{P}$ takes the previous output feature $\boldsymbol{z}_{out,t-1}$ as the input $\boldsymbol{z}_{in,t}$, and the self-attention layers always take all previous inputs $\boldsymbol{z}_{in,0:t}$ to calculate keys and values. After the output feature $\boldsymbol{z}_{out,t}$ is obtained, a linear layer is utilized to convert it into a logit $z_t$. Finally, logits from all time steps are further interpreted into specific attributes in a non-parametric way, according to different tasks. We formulate this process as in Eq. (2).

$$\boldsymbol{z}_{in,t} = \begin{cases} \boldsymbol{f}_o, & t = 0 \\ \boldsymbol{z}_{out,t-1}, & t > 0 \end{cases}$$
$$\boldsymbol{z}_{hidden,t} = SelfAttn(\boldsymbol{z}_{in,t}\boldsymbol{W}_Q; \boldsymbol{z}_{in,1:t}\boldsymbol{W}_{KV})$$
$$\boldsymbol{z}_{out,t} = CrossAttn(\boldsymbol{z}_{hidden,t}; \boldsymbol{F}_I) \tag{2}$$
$$z_t = \boldsymbol{z}_{out,t}\boldsymbol{w}_t^T$$

General Sequence Predictor provides a unified way to generate output attributes for each object. It can handle different object-level visual tasks with an identical structure, which makes Obj2Seq ready to be implemented for general usage.

### 3.4  Other Details

**Image Feature Encoder**   Image Feature Encoder $\mathcal{E}$ consists of a resnet backbone and 6 multi-scale deformable encoder layers, which is exactly the same as [52]. The backbone extracts multi-scale feature maps $\{\boldsymbol{F}_B^{(l)}\}_{l=1}^4$ from $C_3$ to $C_5$ and an additional downsampled scale $C_6$. These feature maps are transformed to have $d = 256$ channels, and then processed by deformable encoder layers to obtain $\boldsymbol{F}_I$.

**Object Transformer Decoder**   Object Transformer Decoder $\mathcal{D}$ consists of 4 multi-scale deformable decoder layers. A decoder layer contains self-attention and multi-scale deformable attention modules, which is elaborated in [52]. Here we only implement 4 decoder layers for fair comparison in computation with other methods.

**Retention policy**   Here we discuss about how Prompted Visual Indicator retains categories and generates class-aware object queries. During training, categories with ground truth objects are first guaranteed to be retained. Then we select categories with the highest scores to fullfill a fixed number of $K'$ classes. While in inference, we can retain either top-$K'$ categories or all categories over a fixed threshold. The influence of these policies is discussed in Section 4.4.3.

**Objectness branch**   Besides describing an object with a sequence, we implement a separate objectness branch to recognize if the object exists. This branch provides a score $s^{(i)}$ for each object as in Eq. (3). We first compute the probability of each object with given image and categories, and then multiply it with its corresponding class existence $s_C^{(k)}$. In Section 4.4.2, we will elaborate on the efficacy of this objectness branch .

$$
\begin{aligned}
s^{(i)} &= P(C|I) \cdot P(O|I, C) \\
&= s_C^{(k)} \cdot sigmoid(linear(\boldsymbol{f}_c^{(k)}) \cdot \boldsymbol{f}_o^{(i)}/\sqrt{d})
\end{aligned}
\tag{3}
$$

**Training Criterion**   The outputs of Obj2Seq is a set of objects with specified categories. These objects are equivalent, and do not appear in a certain order. Therefore, we use bipartite matching loss for supervision [4]. It matches object queries with ground truth so that the loss of matched pairs is minimized, and then optimizes this optimal match. This loss makes sure that each object is only predicted with one object query, eliminating the need of non-maximum suppression. The definition of the detailed loss function varies for tasks, which will be elaborated in Appendix. As to the classifier in Prompted Visual Indicator, we regard it as an auxiliary multi-label classification problem, and supervise it with asymmetric loss [36].

## 4  Experiments

In this section, we first conduct experiments on three different visual tasks separately, including object detection, multi-label classification and human pose estimation. These results demonstrate how Obj2Seq solves visual tasks in a unified way. After that, some ablation studies are provided to better comprehend the detailed options in Obj2Seq. Here we only provide overall results. Implementation details please refer to Appendix.

### 4.1   Experiments on Object Detection

In Table 1, we compare Obj2Seq with a group of popular detection frameworks on MS COCO [28]. We mainly compare it with different variations of DETR [4] here, because they are also end-to-end object detection frameworks without NMS, and Obj2Seq shares some similarity with Deformable DETR [52]. All variations use multi-scale features or dilated convolution in stage 5 (DC5) for similar computation cost. Meanwhile, we also present some representative convolution-based frameworks [35, 40] too, and they are trained with longer schedulers as reported in former papers.

Obj2Seq achieves 45.7% mAP with R50 as the backbone, without bells and whistles. It is +1.2% higher than Deformable DETR using the same multi-scale deformable attention, and outperforms other detection frameworks as well, no matter DETR-like or convolution-based frameworks with a longer scheduler. With R101 as the backbone, Obj2Seq also achieves 46.1%. We notice that

Table 1: Object detection results on MS COCO. All frameworks takes R50 (the upper half) and R101 (the lower half) as the backbone. Results with $^\dagger$ indicates they use a DC5-variation of resnet.

| Method | Epochs | GFlops | Params | $AP$ | $AP_{50}$ | $AP_{75}$ | $AP_S$ | $AP_M$ | $AP_L$ |
|---|---|---|---|---|---|---|---|---|---|
| Faster RCNN+FPN [4] | 108 | 180 | 42M | 42.0 | 62.1 | 45.5 | 26.6 | 45.4 | 53.4 |
| TSP-RCNN [40] | 96 | 188 | 64M | 45.0 | 64.5 | 49.6 | 29.7 | 47.7 | 58.0 |
| Pix2Seq$^\dagger$ [9] | 300 | - | 38M | 43.2 | 61.0 | 46.1 | 26.6 | 47.0 | 58.6 |
| DETR$^\dagger$ [4] | 500 | 187 | 41M | 43.3 | 63.1 | 45.9 | 22.5 | 47.3 | 61.1 |
| SMCA [15] | 50 | 152 | 40M | 43.7 | 63.6 | 47.2 | 24.2 | 47.0 | 60.4 |
| Conditional DETR$^\dagger$ [31] | 50 | 195 | 44M | 43.8 | 64.4 | 46.7 | 24.0 | 47.6 | 60.7 |
| Anchor DETR$^\dagger$ [46] | 50 | 171 | 37M | 44.2 | 64.7 | 47.5 | 24.7 | 48.2 | 60.6 |
| DAB-DETR$^\dagger$ [29] | 50 | 202 | 44M | 44.5 | **65.1** | 47.7 | 25.3 | 48.2 | **62.3** |
| Deformable DETR [52] | 50 | 171 | 40M | 44.5 | 63.5 | 48.8 | 27.1 | 47.6 | 59.6 |
| Obj2Seq-R50 (Ours) | 50 | 184 | 40M | **45.7** | 64.8 | **49.5** | **28.0** | **48.8** | 60.2 |
| Faster RCNN+FPN [4] | 108 | 246 | 60M | 44.0 | 63.9 | 47.8 | 27.2 | 48.1 | 56.0 |
| TSP-RCNN [40] | 96 | 254 | 83M | 46.5 | 66.0 | 51.2 | 29.9 | 49.7 | 59.2 |
| Pix2Seq$^\dagger$ [9] | 300 | - | 57M | 45.0 | 63.2 | 48.6 | 28.2 | 48.9 | 60.4 |
| DETR$^\dagger$ [4] | 500 | 253 | 60M | 44.9 | 64.7 | 47.7 | 23.7 | 49.5 | 62.3 |
| SMCA [15] | 50 | 218 | 58M | 44.4 | 65.2 | 48.0 | 24.3 | 48.5 | 61.0 |
| Conditional DETR$^\dagger$ [31] | 50 | 262 | 63M | 45.0 | 65.5 | 48.4 | 26.1 | 48.9 | 62.8 |
| Anchor DETR$^\dagger$ [46] | 50 | 237 | 56M | 45.1 | 65.7 | 48.8 | 25.8 | 49.4 | 61.6 |
| DAB-DETR$^\dagger$ [29] | 50 | 282 | 63M | 45.8 | **65.9** | 49.3 | 27.0 | 49.8 | **63.8** |
| Obj2Seq-R101 (Ours) | 50 | 251 | 59M | **46.1** | 65.3 | **50.0** | 27.7 | **50.0** | 61.4 |

DAB-DETR achieves significantly higher $AP_L$ than Obj2Seq. This possibly attributes to its enhanced position embeddings, since reference points with learnable $wh$ may better fit objects with extreme scales. In general, these results indicates that Obj2Seq is an effective framework for object detection. It achieves excellent performance.

## 4.2 Experiments on Multi-Label Classfication

Prompted Visual Indicator does not only generate object queries for Object Transformer Decoder, but also classify if the input categories exist in the image, and output scores for them. Therefore, we are able to perform multi-label classification as well with the outputs from Prompted Visual Indicator. We evaluate the model trained in Section 4.1 on MS COCO multi-label benchmark. The classifier in Prompted Visual Indicator achieves 89.0 mAP, which is comparable to most multi-label classification solutions. We will elaborate on its detailed behavior in Appendix.

## 4.3 Experiments on Human Pose Estimation

Current human pose estimation algorithms are generally divided into three categories, bottom-up methods, top-down methods with image crop, and top-down methods in an end-to-end way. All of them involves sophisticate structures, such as image crop [47, 50, 11] or RoI Align [24, 19] for top-down methods and group post-processing for bottom-up methods [33, 3]. There are few works directly obtaining keypoints from the input image [39]. In Obj2Seq, General Sequence Predictor can directly read the coordinates out from object queries without any task-specific model structures. We also provide a baseline for better comparision. The only difference between the baseline and Obj2Seq is that the baseline model use a simple MLP instead of General Sequence Predictor. The details in hyperparameters and training configurations are elaborated in Appendix.

Our experiments are conducted on the COCO keypoint challange [28], and we mainly list other end-to-end top-down methods in Table 2 for comparison. Obj2Seq achieves 60.1% mAP with a 50-epoch training scheduler. It reaches a reasonable performance as an end-to-end framework with no delicate structures. The result with General Sequence Predictor is +2.9% higher than the MLP-head baseline, which indicates that it is a more effective predictor. This improvement may attribute to the interdependence between different keypoints. Knowing the exact position of some keypoints benefits the prediction of others. We also attempt to train Obj2Seq with a longer training schedule. Training

Table 2: Human pose estimation results on MS COCO. We mainly compare with end-to-end frameworks here. Results with $\ddagger$ indicates they utilize a pre-trained detector.

| Method | Backbone | Epochs | $AP$ | $AP_{50}$ | $AP_{75}$ | $AP_M$ | $AP_L$ | $AR$ |
|---|---|---|---|---|---|---|---|---|
| Mask-RCNN [19] | ResNet50 | - | 63.1 | **87.3** | 68.7 | 57.8 | 71.4 | - |
| CenterNet [51] | Hourglass-104 | 150 | 63.0 | 86.8 | 69.6 | 58.9 | 70.4 | - |
| DirectPose [41] | ResNet50 | 100 | 63.1 | 85.6 | 68.8 | 57.7 | 71.3 | - |
| PRTR$^\ddagger$ [24] | HRNet-W48 | - | 64.9 | 87.0 | 71.7 | **60.2** | 72.5 | **74.1** |
| POET [39] | ResNet50 | 250 | 53.6 | 82.2 | 57.6 | 42.5 | 68.1 | 61.4 |
| baseline | ResNet50 | 50 | 57.2 | 83.3 | 63.7 | 51.5 | 66.3 | 65.7 |
| Obj2Seq | ResNet50 | 50 | 60.1 | 83.9 | 66.2 | 54.1 | 69.5 | 68.0 |
| Obj2Seq | ResNet50 | 150 | **65.0** | 86.5 | **71.8** | 59.6 | **74.0** | 72.7 |

Table 3: Experiments on different parts of Obj2Seq.

| Class Prompt | Sequential Predictor | $mAP$ | $AP50$ | $AP75$ | $AP_S$ | $AP_M$ | $AP_L$ |
|---|---|---|---|---|---|---|---|
| | | 44.5 | 63.5 | 48.8 | 27.1 | 47.6 | 59.6 |
| ✓ | | 45.1 | 64.4 | 49.4 | 27.4 | 48.5 | 60.0 |
| ✓ | w/o separate Objectness | 45.1 | 63.8 | 48.9 | 26.8 | 48.5 | 60.0 |
| ✓ | w/ separate Objectness | 45.7 | 64.8 | 49.5 | 28.0 | 48.8 | 60.2 |

for 150 epochs can further improves mAP to 65.0%. Obj2Seq currently falls behind of start-of-the-art methods (listed in Appendix), but has potential for further improvements. Nevertheless, these experiments validate that Obj2Seq can be implemented in human pose estimation, and we believe it is also available for other visual tasks.

## 4.4 Ablation Studies

In this section, we examine different parts and ablate on some important options in Obj2Seq. We conduct these experiments on object detection for demonstration.

### 4.4.1 Overall Structure

We mainly elaborate the existence of two modules in Obj2Seq, Prompted Visual Indicator and General Sequence Predictor. Removing them leading to a framework similar to Deformable DETR. Therefore, we directly takes Deformable DETR as the baseline. Table 3 shows that both modules provide steady improvements for Obj2Seq. With the help of Prompted Visual Indicator, Obj2Seq is able to process objects of different categories separately, and distinguish important class-aware features with given prompts. This makes Obj2Seq work more efficiently, leading to an improvement of +0.6%. General Sequence Predictor is initially designed for better generality. However, the precision is also increased by +0.6%, which actually surprises us. We suppose that attributes of most object-level tasks are not independent. Some known attributes may help with the prediction of the others. For example, it would be more accuracy to estimate the scale of a box if its center is determined.

### 4.4.2 Objectness Branch

In Section 3.4, we have introduced the additional objectness branch. It is constructed in order to recognize if the object exists, rather than to describe it. Meanwhile, it is also a valid way if we merge objectness into the output sequence of General Sequence Predictor, which means inserting an additional word representing the objectness score before the original sequence. We validate the necessity of the objectness branch in Table 3. Predictor without a separate objectness branch achieves the same performance as the original MLP. However, a separate objectness branch further improves mAP by +0.6%. Both options are extensible for different tasks. We explain this phenomenon in two aspects. On one hand, a single linear layer is not that good at judging the existence of objects of different categories. It needs help from the class vectors. On the other hand, checking the existence of an object is different from describing it. Their functions may conflict in the same head. Therefore, it is better to build a separate objectness branch.

Table 4: Experiments on prompt vectors.

| Clip Initialization | Fixed Prompt Vectors | $mAP$ | $AP50$ | $AP75$ | $AP_S$ | $AP_M$ | $AP_L$ |
|---|---|---|---|---|---|---|---|
| | | 45.4 | 64.5 | 49.3 | 27.9 | 48.9 | 60.3 |
| ✓ | | 45.7 | 64.8 | 49.5 | 28.0 | 48.8 | 60.2 |
| ✓ | ✓ | 45.2 | 64.7 | 49.0 | 26.6 | 48.7 | 60.0 |

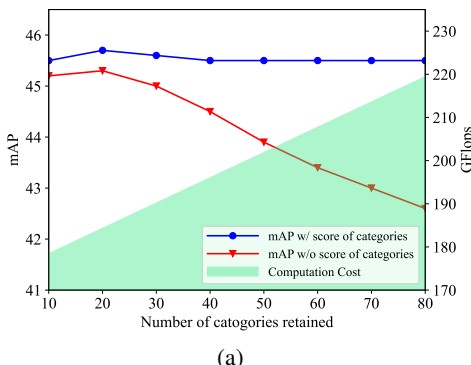
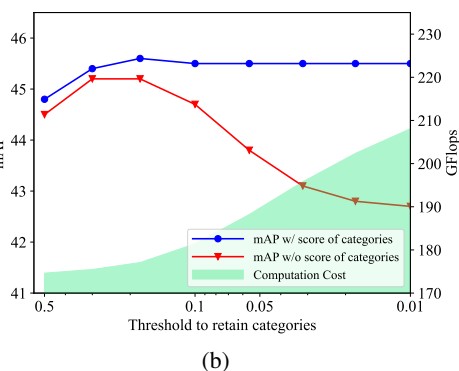

(a)                                                           (b)

Figure 3: We plot how the model behavior changes when different retention policies are utilized in evaluation. The figures are drawn with (a) top-$K'$ policies and (b) fixed threshold policies. The curves demonstrate how mAP fluctuates while the colored areas represents computation cost.

### 4.4.3   Ablation of Prompted Visual Indicator

**Input prompt vector**    Obj2Seq receives class prompts as inputs to detect objects of certain categories, while how to generate class prompts remains a question. The simplest way is learning a prompt vector for each class from scratch, just like how object queries are learned in DETR [4]. As shown in Table 4, this straight-forward solution achieves 45.4%. CLIP [34] is a robust feature extractor pretrained with a huge amount of image-text data pairs. It can provide better representations for label text of each category. Initializing prompt vectors with CLIP features achieves +0.3% improvement. However, fixing these prompt vectors leads to a slight decrease in precision. Vectors generated by CLIP may not be well aligned with image features. It needs to be tuned in the training process.

**Retention Policies**    As mentioned in Section 3.2, Prompted Visual Indicator is not only used for classification, but also to initialize object queries for retained categories. If one category is filtered out, it means that Obj2Seq will not detect any objects of this category. It may be doubtful whether Obj2Seq is sensitive to the retention policy. Here we demonstrate how mAP changes when we use different policies in evaluation. As shown in Figure 3, we mainly test with two policies, retaining the top-$K'$ categories or any categories over a threshold. Obj2Seq achieves steady results with both policies (see the blue curves). The evaluation metric fluctuates within a range of 0.2% when the number of remaining classes changes from 10 to 80, and mAP is also stable when the threshold is lower than 0.3. These results indicate that Obj2Seq is insensitive to the retention policy.

In Eq. (3), the scores for class existence $s_C^{(k)}$ is further multiplied in the objectness branch to obtain final scores for objects. It does not improves the accuracy much if we use the same retention policy in training and evaluation, but makes Obj2Seq a more robust framework with different policies in inference. Figure 3 provides both curves with and without the participance of class existence $s_C^{(k)}$. Without $s_C^{(k)}$, the precision drops more significantly when we modify the retention policy in evaluation. This phenomenon emerges because Object Transformer Decoder and General Sequence Predictor receives inputs in different distributions when the policy changes. Inputs of some categories are filtered out by Prompted Visual Indicator during training, therefore Obj2Seq cannot make accurate predictions if they are seen in inference. By multiplying with the class existence $s_C^{(k)}$, we are able to make it up to some extent.

# 5 Conclusion

In this paper, we propose a controllable and versatile framework for object-level visual tasks, Obj2Seq. This framework takes class prompts to determine the objects it should attend to, and generates a sequence of attributes to describe each object in the image. Obj2Seq can be applied in different visual tasks, such as object detection, human pose estimation and image classification, and achieves comparable results with other task-specific methods.

There are a few problems unsolved in Obj2Seq, which need further discussions. Firstly, Obj2Seq mainly focuses on a unified definition of different visual tasks, but barely pays attention to the detailed structure of the feature extractor. Designing a new feature extractor for general usage may achieve better performance. Secondly, Obj2Seq can be implemented in a wide range of other tasks. Finally, Obj2Seq only takes class prompts to guide the model. In order to satisfy finer requirements, a more detailed indicator is worth developing. All in all, Obj2Seq points a promising direction for general vision models, and it may inspire further studies.

## Acknowledgments and Disclosure of Funding

This work was supported by National Key R&D Program of China under Grant No.2021ZD0110403, National Natural Science Foundation of China (No.62002357, No.61976210, No.62176254, No.61876086, No.62076235), the InnoHK program and also sponsored by Zhejiang Lab (No.2021KH0AB07).

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
