# OpenReview forum: "Obj2Seq: Formatting Objects as Sequences with Class Prompt for Visual Tasks"
_NeurIPS.cc/2022/Conference — NeurIPS 2022 Accept_

### Official Review · Reviewer_BccU · 2022-06-29

**Rating:** 5
**Confidence:** 1
**Soundness:** 2 fair
**Presentation:** 2 fair
**Contribution:** 2 fair

**Summary:**

This paper proposes a new framework for vision tasks, called Obj2Seq. It takes prompts of classes as input then filters the non-exist categories and keeps the most confident K' categories for each image by the Prompted Visual Indicator module. After that, they will be passed to the Object Transformer Decoder module together with the encoded image features to get the object queries. The final General Sequence Predictor takes these queries as input to generate a sequence of desired results, e.g., the location (x, y), size (w, h), or even the other for fine-grained information like keypoints (x_nose, y_nose, x_eye1, y_eye1, .....). The experiments on MS COCO demonstrate that the proposed Obj2Seq can achieve comparable results with other task-specific methods.


**Questions:**

According to the authors, the main contribution of this paper is to provide a unified visual framework, but I only perceive it as an object-level prediction framework, which is equivalent to the conventional Mask R-CNN framework in terms of potential capacity. Therefore, I want to know some specific examples of tasks that can only be formulated by Obj2Seq but not Mask R-CNN during rebuttal.

**Limitations:**

The authors have already addressed the limitations.

**Strengths And Weaknesses:**

[+] This paper proposes a new framework for object-level visual prediction tasks, e.g., object detection, multi-label classification or human pose estimation.
[+] According to the experimental results, the proposed Obj2Seq can achieve comparable performances on each task with their task-specific methods.

[-] However, I don't think Obj2Seq is a unified vision framework for most vision tasks, which is mainly designed to solve the object-level prediction. In my opinion, the capacity of the proposed Obj2Seq is equal to the conventional Mask R-CNN framework, because Mask R-CNN is also able to equip new prediction heads based on the downstream tasks. Therefore, I prefer to consider Obj2Seq as another version of "Mask R-CNN" rather than a unified framework for vision tasks. To better understand what a unified vision framework is supposed to look like, please refer to the [1, 2], which are capable of both localization tasks and understanding tasks. Note that I'm NOT suggesting the authors to compare the proposed Obj2Seq with these methods, as [2] and this paper are contemporary works. What I want to say is that the proposed Obj2Seq is far from a unified vision framework.


[1] Lu, Jiasen, et al. "12-in-1: Multi-task vision and language representation learning." Proceedings of the IEEE/CVF Conference on Computer Vision and Pattern Recognition. 2020.
[2] Zhang, Haotian, et al. "GLIPv2: Unifying Localization and Vision-Language Understanding." arXiv preprint arXiv:2206.05836 (2022).

---

> ### Author Response · Authors · 2022-08-02
> **Response to Reviewer BccU**
>
> We thank the reviewers for detailed comments. Here we elaborate on the Obj2Seq and its difference from Mask R-CNN.
>
> Obj2Seq differs from Mask R-CNN in two aspects. On one hand, Obj2Seq provides a unified output format as sequences for object-level tasks, rather than only a multitask co-training framework. An identical prediction head in Obj2Seq is capable to perform different tasks, without introducing task-specific parameters. While in Mask R-CNN, we need several heads applying to RoIs, each for a certain task. On the other hand, Obj2Seq is able to be assigned with certain categories to obtain desired outputs. After pre-training with large datasets, we can adjust the model behavior during inference according to practical requirements without tuning, which is unreachable for Mask R-CNN. Last but not least, the sequence output and class prompts give us the opportunity and potential to form a unified network structure with NLP, speech, and even multimodality.
>
> However, due to the diverse output formats in computer vision, it is hard to unify image-level, object-level and pixel-level tasks in the same framework. Existing methods like [1,2] still need multiple heads for different vision and NLP tasks, and lack a unified output format. They bear more similarity with a multitask framework. In this paper, we mainly concentrate on object-level tasks, and format them as sequence prediction tasks. We will also generalize towards other visual tasks like pixel-level segmentation in the further research.

---

> > ### Comment · Reviewer_BccU · 2022-08-07
> > **Re: Response to Reviewer BccU**
> >
> > Since this paper is not in my area, my previous score is under-rated. After reading all the reviews, the previous related work like DETR, and author feedback, I decided to raise my score to 5: Borderline accept. However, as also mentioned by Reviewer 7jWi and Reviewer LxeJ, I still think the so-called "unified framework" is a little bit exaggerated. I hope the authors can add some constraints like "for object-level visual tasks" in the later revision.

---

> > > ### Author Response · Authors · 2022-08-09
> > > **Response to Reviewer BccU**
> > >
> > > We thank the reviewers for their careful thoughts and kindly comments. We have updated the manuscript to emphasize more on how Obj2Seq is able to solve object-level visual tasks in a unified way, and therefore make our description more accurate and easier to understand. We will keep working to generalize Obj2Seq to more tasks, and construct a more comprehensive vision framework.

---

### Official Review · Reviewer_tovC · 2022-07-10

**Rating:** 7
**Confidence:** 5
**Soundness:** 4 excellent
**Presentation:** 3 good
**Contribution:** 3 good

**Summary:**

This paper is a nice try towards unified interface for various object related computer vision tasks. It use image and class as prompt, achieving SOTA detection and pose estimation performance in the same interface. The architecture designs of the Prompted Visual Indicator and Sequence Predictor are sound. The major debates lie in the usage of class as prompt and the object-level query. These two might be beneficial to get SOTA object-level tasks, but might also be the blocker for more general vision task interface.

**Questions:**

The limitation of Pix2seq mentioned in this paper that "Pix2Seq is not aware of the desired targets before inference, and the output sequence might become extremely long under sophisticated scenes." This is not convincing enough. We could argue this actually shows the flexibility of Pix2seq. May I bother the author to elaborate more on this? What are the specific scenarios Pix2Seq may meet trouble and Obj2Seq is better.

**Limitations:**

Yes, the author mentioned the limitation of this work.
Class as prompt is not only a limitation to satisfy finer requirements, but also cause trouble on open-vocabulary applications.

**Strengths And Weaknesses:**

Strengths:
1. This effort addresses an ambitious goal of getting a unified interface for computer vision tasks. Obj2seq unifies object detection, pose estimation and image classification.
2. The designed architecture achieves SOTA performance on object detection and pose estimation.

Weaknesses:
1. Class as prompt blocks its way to generalize to open-vocabulary applications. The goal of the unified interface is to utilize the scaling law of large scale model to utilize large dataset and address real-world open-vocabulary challenges.
Debatable:
1. The architecture keeps the object query, while pix2seq has already gotten rid of that. This is debatable. I don't have an obvious answer whether that part is necessary to achieve SOTA performance on several object-level tasks considering object is the unit that interaction or reasoning works on.

---

> ### Author Response · Authors · 2022-08-02
> **Response to Reviewer tovC**
>
> We thank the reviewers for detailed comments. Here we elaborate on why the prompt indicator and object queries will not block further extensions.
>
> **The capability for new class discovery and open-world challenges**
>
> Instead of blocking the way to open-vocabulary applications, class prompts can help large-dataset training and real-world applications with existing text embeddings (e.g., BERT, CLIP). These embedding models are trained with large-scale text or image-text data. They can not only cover new categories, but also improve model performance with embedded semantic knowledge. In Table 4, we demonstrate CLIP embeddings can be directly taken as prompts to represent input categories. Therefore, it is highly flexible to modify the input prompts in practical utilities, and scale the label space and training data to obtain a large general model. Moreover, the modification made in some class prompts would not influence the results of other categories, since all prompts are processed independently in Obj2Seq.
>
> **Object queries provides a structured and unified solution for object-level tasks**
>
> A unified sequence prediction task is a promising direction to unify vision and NLP tasks, while high-dimensional outputs of sophisticated tasks can hardly be represented by a single sequence. Pix2Seq succeeds with object detection [1], but fails to generalize over keypoint detection and instance segmentation in [2]. It can only perform these tasks on object-centric crops. In object-related tasks, objects are appropriate units to organize features and output information. Therefore, we borrow the concept of object queries to better adapt Obj2Seq for object-level visual tasks, and formulate these tasks as sequence generation tasks for each object. On one hand, Obj2Seq provides a more structured output to better enhance object-level task performance. On the other hand, these tasks take a unified format so that they can be handled by an identical framework in an end-to-end manner. We will keep working to generalize Obj2Seq over other image-level and pix-level tasks, to reach the goal of a unified interface for all visual tasks in the future.
>
> **Some practical problems about Pix2Seq**
>
> Pix2Seq [1,2] points out a promising direction to formulate visual tasks as sequence generation problems like NLP. However, it meets two problems. Firstly, Pix2Seq is not able to take additional knowledge as prompts, such as existing language priors pretrained on large-scale text data. Since Pix2Seq always generates a fixed object set, we are not able to use external knowledge to guide the detection of changeable categories. Therefore, it has limited ability in open-vocabulary detection. On the contrary, Obj2Seq can take CLIP embeddings as prompts, and output objects within an arbitrary category set. Secondly, introducing more tasks like keypoint detection and instance segmentation would result in even longer sequences and higher computation cost. In order to perform these tasks, Pix2Seq v2 crops each objects out in a compromised way. By contrast, Obj2Seq generates sequences for all objects in parallel, and performs these tasks in an end-to-end way.
>
> [1] Pix2seq: A language modeling framework for object detection, ICLR 2022.
>
> [2] A Unified Sequence Interface for Vision Tasks, arXiv 2206.07669.

---

> > ### Comment · Reviewer_tovC · 2022-08-06
> > **Thanks for the response**
> >
> > Thanks for the response. I raised the score.
> >
> > I like the discussion in the author response. I tend to prefer the paper to be accepted so that broader readers can discuss more on the topics of:
> > 1. Open-world tasks utilizing multimodal pretraining;
> > 2. Would object query architecture be necessary to get better performance for some object-level tasks? The discussion would be interesting regarding some recent related works as pix2seq v1/v2.

---

> > > ### Author Response · Authors · 2022-08-06
> > > **Response**
> > >
> > > We appreciate the reviewers' kindly and thoughtful comments, these inspire us a lot. We would continue digging into these topics, and try our best to make contributions in the field of a unified vision interface.

---

### Official Review · Reviewer_LxeJ · 2022-07-11

**Rating:** 5
**Confidence:** 3
**Soundness:** 2 fair
**Presentation:** 3 good
**Contribution:** 2 fair

**Summary:**

The paper introduces a unified architecture similar to M-DETR, but takes object class labels as input, and outputs three components in the same architecture:
1. the existence of object for multi-label classification
2. the box coordinates for object detection
3. keypoint coordinates for human pose estimation.

The key component of this unified architecture is the sequence decoder, from which the first four outputs are always trained to be box coordinates and the rest outputs are trained to be human pose keypoints.


**Questions:**

1. Are the Figure (3) a & b actually the same figure? I.e. the x-axes have one-to-one mapping and can be plotted in the same figure.
2. What if there are multiple objects of the same category in the image? From Figure 2 it looks like there are 3 object queries l for each class embedding output. Does it work if the image has more than 3 people? Could the authors add more details about how to handle these multi-instance cases?


**Limitations:**

The paper discusses three limitations: (1) not exploring the design of the feature extractor, (2) not exploring other tasks, (3) only taking object class as input and not exploring more detailed indicators. I agree with these limitations as potential future works, but I think point (2) has a higher priority given the paper claims a "unified framework".

Additionally, I think one limitation of the model is that the object categories are limited by the input categories, and the model cannot discover new categories.


**Strengths And Weaknesses:**

The novelty of the paper is limited. The paper claims a "unified framework" (e.g. Figure 1) and "a wide range of vision tasks" (L91) but the experiments only show the tasks of outputting spatial coordinates (box coordinate and keypoint). While some recent work OFA (I’m aware it was not officially published at the submission time) shows a seq2seq framework can be applied to a much wider range of tasks including detection, caption and VQA. Overall I mean more vision tasks are required to support the claim of a “unified framework”. Second, the authors design an architecture that is similar to DETR-family, and the previous work M-DETR uses the architecture for language grounding tasks. This paper takes the class labels as input and the model outputs the bounding box, which feels more like a simplified version of a visual grounding task rather than object detection. Can the authors elaborate more on the difference between the proposed setting and visual grounding tasks?

The overall quality of the paper is good. The experiments are clearly explained and discussed. However, I think the comparison with previous works (Table 1) is unfair since the method uses a retention policy to pre-filter some probable object categories while the normal detection method does not benefit from this. Can the authors discuss this comparison more?

In general, the paper is very clearly written and well organized. The figures are clear and easy to understand.

The experiments in object detection show the proposed method outperforms previous methods but due to the retention policy (mentioned above), I doubt the fairness of this comparison and more discussion is required. The experiments in human pose estimation show the proposed method is effective when compared with previous works.

---

> ### Author Response · Authors · 2022-08-02
> **Response to Reviewer LxeJ**
>
> We thank the reviewers for detailed comments. Here we address them separately:
>
> **How Obj2Seq generalizes over object-level tasks**
>
> We intend to build a unified framework for object-level visual tasks. Current methods like OFA mainly conduct experiments on tasks only generally describing an image (e.g., classification and text-image matching). Even though they utilize detection datasets for pretraining, OFA does not provide evaluation metrics on object detection. Therefore, it is still far from unifying visual tasks with various output formats. Among visual tasks, object-level outputs usually contain sophisticated and high-dimensional information which can hardly fit into a single sequence. Obj2Seq concentrates on these tasks. We format these tasks as sequence generation problems for each single object. This works as a unified interface for object-level tasks. We take object detection and human pose estimation as examples to demonstrate its efficacy. When new tasks are encountered, Obj2Seq adapts by modifying the definition of the output sequences only. We will keep supporting more tasks with Obj2Seq, such as pixel-level segmentation, and release our code for public.
>
> **Difference from DETR-family and visual grounding tasks**
>
> Obj2Seq works as a unified interface for various object-level visual tasks, and borrows the idea of object query from DETR-family and M-DETR to extract better object-related features. However, by predicting sequences as outputs, Obj2Seq is able to generate outputs for various tasks, more than just object detection. As to visual grounding, our problem set is different in two aspects. On one hand, visual grounding takes a detailed description as the input, and identifies only one specific object in the input image for each description, while Obj2Seq detects all objects belonging to the input categories. On the other hand, visual grounding only outputs the bounding box of the referred object, while Obj2Seq is able to generate any required description in an output sequence.
>
> When evaluating on COCO, Obj2Seq takes a fixed set of 80 categories as input prompts for fair comparison with other detection frameworks. However, Prompted Visual Indicator leaves us an interface to interfere the model inference with changing category group, thus satisfying varied real-world requirements. This idea is somewhat inspired by visual grounding tasks, and we will further generalize this visual indicator with more detailed text data.
>
> **Fairness about the retention policy**
>
> The retention policy has minor impact on the model performance. As the blue line in Figure 3(a), there is minor difference between the accuracies with the best policy (45.7, $K'=20$) and without a retention policy (45.5, $K'=80$, i.e., remain all categories). Both of them perform better than previous works in Table 1.
>
> Obj2Seq achieves improvements in accuracy from two aspects. The first is that class prompts help object queries focus on corresponding categories. The second is that a sequence structure can better perceive relations among different output steps. These have been validated in Table 3.
>
> **Implementation details about object queries**
>
> Obj2Seq generates object queries based on the input image and categories. It first filters and maintains several top categories that mostly likely exist (assuming $K'$ categories), and then generates $N$ object queries for each retained category ($K'N$ queries in total). In our experiments, we set $N=100$. For more detailed configurations, please refer Appendix A.1.
>
> **Clarification of variables in Figure 3**
>
> These two variables, the number of remained categories and the threshold to remain, do not have a fixed mapping relationship. If one is fixed, the other changes with input images. In Figure 3(a) and 3(b), we demonstrate Obj2Seq is able to achieves steady performance no matter we control the retention policy by number or threshold.
>
> **The capability for new class discovery and open-world challenges**
>
> In Table 4, we validate that Obj2Seq can also directly use CLIP embeddings as input prompts, which makes it capable for large-dataset and real-world open-vocabulary challenges. Prompt indicator is designed to assign this general model some specific categories according to practical requirements. The input prompts can be changed flexibly, and Obj2Seq then generates corresponding output sequences. We will also extend the indicator to further support more various and detailed prompts and combine it with text data.
>
> More specifically, though experiments on COCO takes a fixed set of 80 categories, each category is processed independently in Obj2Seq. We can flexibly modify some prompts without influencing others.

---

> > ### Comment · Reviewer_LxeJ · 2022-08-08
> > **Reply to authors**
> >
> > Thanks for the response, it clarifies most of my concerns.
> >
> > A minor point about Figure 3, I still think Figures 3(a) and 3(b) can share the same x-axis, if the metric for selecting top-K is the same metric for thresholding, which I believe it is. I don't understand why "if one is fixed, the other changes with input images", can I bother the authors to explain a bit more?
> >
> > Overall, I slightly raise my score from "borderline reject" to "borderline accept". The main reason is similar to reviewer `tovC`'s point, that the paper could bring some discussions around the object-centred architectures and open-world vocabulary tasks. However, I suggest claiming the "unified framework" carefully or adding some constraints like "object-centric".

---

> > > ### Author Response · Authors · 2022-08-09
> > > **Response to Reviewer LxeJ**
> > >
> > > We thank the reviewers for their careful thoughts and kindly comments.
> > >
> > > If we take a fixed threshold to retain categories, the number of categories above this threshold vary for different input images. Some images may have tens of categories retained, while some have few. This is because the scores of class existence are calculated based on each input image, and independent for each category. They do not have a fixed distribution. Similarly, when we take the top-$K$ policy, we retain a fixed number of $K$ categories, even if some of them have a relative low score.
> > >
> > > Meanwhile, we have modified the description in our manuscript, and emphasize that Obj2Seq is a unified framework for object-level visual tasks. We will keep working to generalize this framework towards a wider range of other visual tasks, and construct a more unified framework in the future.

---

### Official Review · Reviewer_7jWi · 2022-07-11

**Rating:** 6
**Confidence:** 5
**Soundness:** 4 excellent
**Presentation:** 3 good
**Contribution:** 3 good

**Summary:**

This paper presents a new object-centered framework to unify visual tasks including object detection, key point detection, and multi-label classification. Different from the previous Pix2Seq method, this paper divides the sequence generation problem into an object query generation sub-problem and an attribute sequence generation sub-problem. The method achieves good results on multiple visual recognition tasks on COCO.

**Questions:**

Overall, I think this is a good paper with several new and insightful technical contributions and decent results. Although I still have concerns about some designs, I think the proposed object-centered framework might be a new and interesting direction for several visual recognition tasks. Therefore, I lean toward accepting this paper. This paper can be stronger if the questions mentioned in the Weaknesses subsection  can be addressed.

**Limitations:**

The limitations and potential societal impact of the method have been discussed.

**Strengths And Weaknesses:**

Strengths:

- The idea of decoupling the sequence generation process into two steps is new and insightful.

- The object query generation process allows the use of bipartite matching loss for supervision, which improves the optimization of the original pix2seq and helps the framework achieve competitive results with state-of-the-art methods.

-  The method is evaluated on multiple popular benchmarks and achieves good results. Ablation studies in Section 4.4 make the paper solid.

Weaknesses:

- Since the method uses bipartite matching loss from DETR [4] for supervision and generates many redundant queries (100 queries for each candidate category according to Appendix A) like [4], I think the overall framework can be viewed as an improved DETR with class conditioned query generator, which is close to several previous methods like [28, 43, 26, 49]. The object-centered design also makes the method less general compared to the concurrent work pix2seq v2 [r1] which can be extended to the image captioning task.

-  The two-step design makes the necessity of the sequence generation framework questionable. The sequence generation task in Pix2Seq is design to unify vision and NLP tasks. However, due to the existence of the Prompted Visual Indicator and bipartite matching loss, the proposed method is totally different from sequence-to-sequence autoregression framework in NLP. Therefore, is that possible to further simplify the General Sequence Predictor as a 2-token transformers model where the object query is combined with detection and keypoint task tokens as the inputs and use MLP heads like DETR to obtain the final prediction?

- The paper emphasizes a unified framework is designed for various visual tasks. However, the experiments didn't show the potential of the proposed framework to perform multiple challenging tasks like [r1]. Is it possible for the proposed framework to perform general object detection and keypoint detection with a single model? Will the model benefit from joint training?

[r1] A Unified Sequence Interface for Vision Tasks, arXiv 2206.07669

---

> ### Author Response · Authors · 2022-08-02
> **Response to Reviewer 7jWi**
>
> We thank the reviewers for detailed comments. Here we address them separately:
>
> **The design of object queries, and how Obj2Seq generalizes over object-level tasks**
>
> Obj2Seq intends to construct a unified framework for object-level tasks. These tasks usually require highly-structured outputs. Therefore, we adopt object queries and bipartite matching loss, in order to extract better object-related features and achieves SOTA results. These designs are inspired by DETR. However, Obj2Seq unifies the output format of each object as a sequence, which is ready to generalize to different object-level tasks. Meanwhile, Obj2Seq performs all object-level tasks in an end-to-end way, which is more simple and friendly than previous unified framework, Pix2Seq v2. The latter needs to crop each object out first to conduct sophisticated tasks in a compromised way. Currently, Obj2Seq is mainly specialized in object-level tasks. We will keep working to generalize it towards other tasks, such as pixel-level segmentation.
>
> **Necessity of the sequence generation framework under the two-step design**
>
> The sequence prediction task is a promising direction to unify vision and NLP tasks. Currently, the main obstacle lies in the difficulty to format diverse visual task outputs in an identical sequence. Instead, Obj2Seq adapts the original auto-regression framework for object-level tasks. It formats each object as a sequence, which provides a more structured and unified interface. On one hand, a sequence is a general format to describe an object. It can be adapted for different tasks. On the other hand, organizing the outputs by objects provides more explicit supervision, leading to better performance. The experiments on object detection and human pose estimation validate that Obj2Seq is capable to generalize over object-level tasks, and achieve SOTA results. Ablation study in Table 3 further indicates that the sequence format can even lead to better performance. This object-sequence interface is ready to be extended to other object-level tasks, and we are going to further adapt it for other image-level and pixel-level tasks.
>
> As to the option of a 2-token transformer, it is much simpler for detection and keypoint only. However, adding new tasks requires introducing more query tokens and new MLPs, which makes it less friendly to general extensions.
>
> **The potential to perform multiple tasks with a unified framework**
>
> The inference pipeline of Obj2Seq keeps unchanged for different object-level tasks, while Pix2Seq needs to crop objects out for tasks like keypoint detection. Our identical and end-to-end pipeline makes Obj2Seq more friendly for new task extension and multi-task training. When performing different tasks, we only need to change required prompt inputs and interpret output sequences according to specific tasks. For example, we can train people detection and keypoint together, and achieves 57.3 mAP and 65.0 mAP.
>
> Currently, we are trying to combine general object detection and person keypoint. By simply combining a batch for each task together, we obtain 44.7 mAP on detection and 54.2 mAP on keypoint. Since all objects have bounding box, but only people have keypoint annotations, and their data amounts differ a lot, the training configuration requires delicate tuning to balance different tasks. We believe after tuning a single unified model is able to achieve comparable performance with single-task models.

---

> > ### Comment · Reviewer_7jWi · 2022-08-07
> > **Thanks for the response**
> >
> > I would like to thank the authors for the detailed response. The response addressed part of my concerns, but I still have some questions about the sequence generation framework:
> >
> > - I am still not convinced that the sequence interface can better generalize to other tasks. Since the sequence interface still needs extra task embeddings to indicate desired tasks, I think the 2-token transformers can also be extended to other tasks with additional task embeddings.
> >
> > - I still think the 2-token transformer is a very important baseline for the proposed sequence interface. It would be better to provide to the results of the 2-token transformer, which can be a useful reference for future research in this direction.

---

> > > ### Author Response · Authors · 2022-08-09
> > > **More about 2-token transformer head**
> > >
> > > We thank the reviewers for their careful thoughts and kindly comments.
> > >
> > > A 2-token transformer requires delicate design for different tasks (e.g., 4-d MLP layers for detection and 34-d MLP layers for keypoint). While in Obj2Seq, we utilize a unified sequence head without task-specific parameters. On one hand, task embeddings all share an identical format. The model behavior is able to keep unchanged for different task embeddings. On the other hand, the task embeddings can be further combined with pre-trained text feature in NLP to eliminate finetuning for new tasks in the future.
> > >
> > > Here we also implement the 2-token transformer head on human detection and keypoint. With the help of task tokens, 2-token head achieves higher performance than the simple MLP baseline. However, since Obj2Seq takes definite outputs from previous steps and utilizes them as inputs for subsequent steps, it is able to capture more explicit intra- and inter-task relations. Therefore, Obj2Seq achieves even better results. Moreover, this unified sequence format is consistent with text and audio tasks. It is more friendly to be extended for other multi-model applications. We have updated these results in the supplementary material.
> > >
> > > | Experiment | Epochs | $AP_{det}$ | $AP_{det}^{50}$ | $AP_{det}^{75}$ | $AP_{kps}$ | $AP_{kps}^{50}$ | $AP_{kps}^{75}$ |
> > > |:---:|:---:|:---:|:---:|:---:|:---:|:---:|:---:|
> > > | Baseline   | 50     |   53.7 | 78.6 | 58.9 | 57.2 | 83.3 | 63.7 |
> > > | 2-Token    | 50     |   53.9 | 80.2 | 58.4 | 58.3 | 83.7 | 64.9 |
> > > | **Obj2Seq**| 50     |   **54.4** | **80.3** | **59.4** | **60.5** | **83.9** | **67.3** |

---

### Meta-Review · Area_Chair_cAGT · 2022-08-28

**Recommendation:** Accept
**Confidence:** Certain

**Metareview:**

The paper proposes an approach for formulating a few visual tasks as sequence prediction with class prompt. Reviewers are overall positive about the paper, especially the direction towards a unified vision model where the paper is exploring. However, it is also pointed out the paper should be more explicit about how the sequence is modeled with object queries and bipartite graph matching loss, which are significant differences from standard sequence modeling, as presented in language models, or Pix2Seq v1/v2. The authors should consider point out these differences in the abstract, Figure 1/2, to avoid misleading readers into thinking this is just like language modeling with autoregressive loss. Overall I’d recommend accepting the paper given it is a good attempt towards a unified vision model, but also encourage the authors to further improve the writing and clarify the sequence modeling part as mentioned above.


**Award:**

No

---

### Decision · Program_Chairs · 2022-09-14

Accept